# Roles of Aerotolerance, Biofilm Formation, and Viable but Non-Culturable State in the Survival of *Campylobacter jejuni* in Poultry Processing Environments

**DOI:** 10.3390/microorganisms10112165

**Published:** 2022-10-31

**Authors:** Diksha Pokhrel, Hudson T. Thames, Li Zhang, Thu T. N. Dinh, Wes Schilling, Shecoya B. White, Reshma Ramachandran, Anuraj Theradiyil Sukumaran

**Affiliations:** 1Department of Poultry Science, Mississippi State University, Mississippi, MS 39762, USA; 2Tyson Foods, 2200 W. Don Tyson Parkway, Springdale, AR 72762, USA; 3Department of Food Science, Nutrition, and Health Promotion, Mississippi State University, Starkville, MS 39762, USA

**Keywords:** *Campylobacter*, environmental stressors, resistance, aerotolerance, biofilm, VBNC

## Abstract

*Campylobacter jejuni* is one of the most common causes of foodborne human gastroenteritis in the developed world. This bacterium colonizes in the ceca of chickens, spreads throughout the poultry production chain, and contaminates poultry products. Despite numerous on farm intervention strategies and developments in post-harvest antimicrobial treatments, *C. jejuni* is frequently detected on broiler meat products. This indicates that *C. jejuni* is evolving over time to overcome the stresses/interventions that are present throughout poultry production and processing. The development of aerotolerance has been reported to be a major survival strategy used by *C. jejuni* in high oxygen environments. Recent studies have indicated that *C. jejuni* can enter a viable but non-culturable (VBNC) state or develop biofilm in response to environmental stressors such as refrigeration and freezing stress and aerobic stress. This review provides an overview of different stressors that *C. jejuni* are exposed to throughout the poultry production chain and the genotypic and phenotypic survival mechanisms, with special attention to aerotolerance, biofilm formation, and development of the VBNC state.

## 1. Introduction

In the United States, it is estimated that *Campylobacter* is responsible for 1.5 million illnesses each year [1] and is one of the most common causes of foodborne gastroenteritis in humans [2]. Consumption of contaminated poultry meat is the primary mode of transmission for *Campylobacter* and is estimated that poultry is responsible for up to 30% of human *Campylobacter* infections [3,4]. Although there are 16 species of *Campylobacter*, *C. jejuni* is responsible for the majority of human infections [3]. Campylobacteriosis is characterized by symptoms such as diarrhea, cramps, fever, and vomiting and has been associated as a major predisposing causes of Guillain-Barre syndrome, a nervous system disorder affecting peripheral nerves [5]. *Campylobacter* is a microaerophilic pathogen (requires 5–10% oxygen) with optimum growth at 42 °C [6,7]. Historically, it has been found that *Campylobacter* lacks the mechanisms present in other bacteria to remove reactive oxygen species (ROS) to defend against oxidative stress. Therefore, they are highly sensitive to environmental oxygen [8]. Despite these physiological restrictions and numerous pre-and post-harvest intervention strategies during poultry processing, *C. jejuni* is frequently detected in retail broiler products [9]. *C. jejuni* tend to survive harsh environmental conditions in poultry processing facilities, which include an aerobic environment, hot water treatments (scalding), chilling and freezing, and usage of antimicrobials such as chlorine [10]. The evolution of survival strategies against these stressors makes it difficult to eliminate *C. jejuni* from food processing surfaces.

One of the major reported mechanisms that enable *Campylobacter* to survive in poultry production and processing environments is the development of aerotolerance. Despite requiring microaerophilic conditions for survival, *Campylobacter* has been reported to exhibit tolerance to oxidative stress [11]. Oxidative stress arises when ROS levels in bacteria surpass the cell’s ability to protect itself. Given that the number of human intestinal campylobacteriosis cases is increasing, it is reasonable to assume that *Campylobacter* has adapted to survive oxidative stress across the food chain. In fact, a high aerotolerance has been reported in *Campylobacter* strains involved in human infections [12,13]. Out of 121 *C. jejuni* strains studied, it was reported that 65 were hyper-aerotolerant (HAT) and 46 were moderately aerotolerant (IAT) [13]. Similarly, High occurrence (63%) of HAT (able to survive more than 24 h of aerobic exposure) isolates in broiler carcass, dairy products, and clinical samples have also been observed, rather than aero-sensitive strains (AS) [14]. Historically, the environmental survivability of *C. jejuni* has been underestimated because of its specific growth requirements and environmental sensitivity. However, the recent reports of aerotolerant strains suggests that increased monitoring of this pathogen is essential in poultry production and processing.

One of the key survival mechanisms of *Campylobacter* may be biofilms. Biofilm formation is the mechanism that various bacteria use to survive environmental stress. Biofilms are defined as ‘multicellular communities of bacteria embedded within a matrix of extracellular polymeric substances that consist of various proteins, polysaccharides, and water’ [15]. Bacteria in biofilms exhibit greater resistance to external stressors than their planktonic counterparts [16]. Recent studies have shown that *C. jejuni* can form biofilm on various abiotic surfaces like acrylonitrile butadiene styrene and polyvinyl chloride plastics commonly used in watering systems and stainless steel commonly used in processing facility [17,18,19]. In food processing facilities, the formation of biofilms may protect *Campylobacter* from cleaning and disinfection protocols and facilitates continued dissemination. This leads to prolonged cross-contamination of food products, thus increasing its potential to cause disease [20]. A particularly concerning aspect of the biofilm formation of *Campylobacter* is its increased survivability at low temperatures [21]. Prolonged survival of *C. jejuni* biofilm at refrigeration temperatures (4 °C and 10 °C) was observed by Dykes et al. [22]. These results suggest that temperature is an important factor that impacts the ability of *Campylobacter* to form biofilms. Similarly, biofilm formation is also accelerated when food surfaces or organic substances are involved. *C. jejuni* biofilm formation was accelerated in brucella broth with chicken meat exudate on glass and polystyrene surfaces [19]. However, compared to other foodborne pathogens such as *Salmonella*, biofilm forming attributes of *Campylobacter* are still unclear in terms of genetic determinants.

Viable but non culturable state (VBNC) is another mechanisms used by many bacteria in response to environmental stressors such as nutrient starvation, osmotic shock, temperature, and pH fluctuations [23,24,25]. In the VBNC state, the ability to culture bacteria is lost even though the bacteria is alive and metabolically active [26]. This state allows bacteria to survive harsh conditions until the environment becomes favorable for growth and cell division [27]. In poultry processing facilities, the bacteria face post-harvest stress and may enter a VBNC state, thereby making their detection difficult during quality control inspection. The ability of *C. jejuni* to enter a VBNC state was first described in 1986 by Rollins and Colwell [28]. Studies have been reported that *C. jejuni* can enter the VBNC state in the presence of acid or cold stress [23,29]. However, the VBNC forming ability of *C. jejuni* and its ability to cause infection in humans is still not clear. *C. jeuni* ability to become non-culturable when exposed to stresses such as heat stress, osmotic stress, acid stress, and refrigeration in the processing facility, as well as its potential to revive and cause illness when consumed by people, needs to be investigated.

The purpose of this review is to summarize the stressors that *C. jejuni* is exposed to in poultry processing environment and to critically analyze the role of various survival mechanisms that are used by this pathogen to overcome the stressors. Brief descriptions of the specific stressors encountered, the potential regions of exposure in the processing plant, and the survival mechanisms used under the given circumstances, have been provided. Potential genes implicated in survival under various stress scenarios are also listed. Understanding the effect of each stressor and the survival processes adopted by this pathogen in overcoming stressors may aid in identifying future avenues for building effective *C. jejuni* control measures in the food chain.

## 2. Physiology of *Campylobacter*

*Campylobacter* requires microaerobic conditions and an optimum temperature of 37–42 °C to grow [30]. The term “microaerobic” refers to oxygen (O_2_) concentrations ranging from 3 to 15%. Although most *Campylobacter* strains are cultured with about 10% oxygen, the usual environment for *C. jejuni*/*C. coli* isolation comprises of approximately 5% oxygen, 10% carbon dioxide (CO_2_), and 85% nitrogen (N_2_) [31].

*Campylobacter* spp. are helical bacteria that may change shape to become rod or coccoid when exposed to specific environmental factors [32]. Spiral shapes are more common in young cultures, but coccoid patterns are more common in older ones [33,34]. The older cells of *C. jejuni*, according to some researchers, are degenerated to the coccoid form and resemble similar coccoid forms that have been observed in several chemoheterotrophic spirilla species [35]. Growing *C. jejuni* in liquid culture resulted in gross morphological changes over time, and that long culture incubation (late decline phase of growth curve) resulted in the formation of a coccal population [36]. In fact, *C. jejuni* becomes coccoid due to an increase in peptidoglycan dipeptides and a reduction in tri- and tetrapeptides. The hydrolases DL-carboxypeptidase Pgp1 and LD-carboxypeptidase Pgp2 separate tri and tetrapeptides and remodel the peptide glycan (PG) structure, and the loss of any of these hydrolases affects the structure and form of the PG, thus compromising the stress tolerance of *C. jejuni* [37].

Further, *Campylobacter* species are unique from other foodborne pathogens in that they have a particular appearance, physiological needs, and a small genome. The *C. jejuni* (NCTC11168 strain) genome is 1.6 Mbp in size, with a low number of repetitive sequences and no insertion or phage-related regions. Interestingly, it also it lacks the antioxidant stress defense regulators SoxRS and OxyR, which are found in *Salmonella* and *E. coli* [38]. In a similar manner, *C. jejuni* possesses only one superoxide dismutase gene (*sodB*), unlike *E. coli*, which possesses three of these genes (*sodA*, *sodB* and *sodC*), respectively [39]. *C. jejuni* has just one subunit of alkyl hydroperoxide reductase (*AhpC*), in contrast to *Salmonella* Enterica, which has two subunits of alkyl hydroperoxide reductase [40]. The function of these genes and their role in overcoming stress has been discussed below.

## 3. Incidence of *Campylobacter* in Poultry Processing Facilities

*Campylobacter* is typically introduced into a manufacturing facility via birds (up to 10^9^ cells/g cecal material) [41]. *Campylobacter* is a commensal organism of many avian species and may be traced back to the farm despite numerous on-farm treatments. It can spread to poultry meat after it reaches the slaughter line, especially during defeathering and evisceration [42]. Poultry meat becomes contaminated with *Campylobacter* due to intestinal breakage, loss of feces from the cloaca, and contact with contaminated equipment, water, or other carcasses. The birds are unloaded, shackled, slaughtered, scalded, defeathered, eviscerated, washed, cooled, and packaged inside a processing facility. The cross contact between the birds, employees, and equipment during these processing steps could contribute to *Campylobacter* contamination. A number of reports disclose the incidence of *C. jejuni* at various stages of processing which can be seen in Table 1.

The persistence of *Campylobacter* in the plant environment and its significance as a source of contamination are unclear. A *Campylobacter* positive flock that arrives at a processing plant can contaminate successive batches of birds that arrive at the same processing plant and are slaughtered prior to the cleaning and sanitation shift at the plant [43,44]. In addition, despite thorough cleaning and disinfection, *C. jejuni* can survive overnight on food processing equipment surfaces [45]. These findings indicate that some *C. jejuni* isolates may be able to survive longer in the processing plant, which could infect successive batches of carcasses from flocks that are not positive for *Campylobacter*.

## 4. Stress Encountered by *Campylobacter* inside Poultry Processing Facilities

Immersion of broiler carcasses in hot water (scalding), rapid cooling (chilling), use of anti-microbials (spraying/dipping), and active packaging (modified atmospheric condition) are some of the hurdles (Figure 1) that *Campylobacter* is subjected to inside the processing facility. Detailed explanations of *C. jejuni* response to each of these stress condition has been discussed below. Involvement of genes for *Campylobacter* to overcome these stressors has been highlighted in Table 2.

### 4.1. Aerobic Stress

The primary stress that *C. jejuni* faces as a microaerophilic bacterium is the oxygen level in the surrounding environment. Presence of oxygen can lead to the formation and accumulation of a wide range of hazardous intermediate products, which are known as ROS including the superoxide anion radical (O_2_), hydrogen peroxide (H_2_O_2_), and the hydroxyl radical (OH) [46]. When ROS levels in bacteria reach a level that exceeds the cell’s ability to defend itself, oxidative stress occurs. Despite its susceptibility to oxidative stress, the rising prevalence of human intestinal campylobacteriosis suggests that *Campylobacter* has adapted to withstand oxidative stress across the entire food chain.

Recent studies iterate that HAT strains of *C. jejuni* are common in human infections that have been linked to poultry meat [12,13,14] *C. jejuni* strains from both retail chicken and duck meat were found tolerant to aerobic stress. [47]. High occurrence (63%) of HAT *C. jejuni* isolates in broiler carcass, dairy products, and clinical samples have been observed rather than aero-sensitive strains (AS) [14]. In fact, several parameters have been linked to *C. jejuni* survival in oxygen-rich environments. For example, pyruvate has been shown to protect *C. jejuni* from severe oxidative stress in aerobic circumstances and a combination of ferrous sulfate, sodium metabisulfite, and sodium pyruvate promotes *C. jejuni* viability in culture media [11]. Furthermore, metabolic commensalism with *Pseudomonas* spp. in the rotting microbiota of chicken flesh has been shown to support *C. jejuni* survival under aerobic conditions [48].

### 4.2. Heat Stress

The ideal temperature for growth of *Campylobacter spp* is between 42 °C and 45 °C. It has been observed that below the minimal growth temperature (31 °C), there is a sudden decrease in the growth rate of *C. jejuni* [49]. In poultry processing plants, scalding occurs at approximately 53 °C to 62 °C in order to loosen feather and aid in their removal. While these temperatures would normally hinder *Campylobacter* growth, some clinical strains, which are tolerant to high temperature pasteurization for both short time (72 °C for 15 S) and long time (72 °C for 30 S) exposure, would not be affected by these scalding temperatures [13]. In addition, the ability to survive heat stress varies with genotypic diversity. For example, *C. jejuni* assigned to clonal complex CC-21 indicated better survivability after exposure to heat stress when compared to CC-45 when the heat stress inside a broiler processing facility was mimicked in the study [50].

Despite having an optimum temperature requirement for growth, *Campylobacter* appears to be resistant to high temperatures, and several studies have been conducted to understand the mechanism of heat stress survival in *Campylobacter* [49,50,51]. In one study, twenty four proteins were more abundant in *Campylobacter* when cells were heat-shocked at temperatures ranging from 43 °C to 48 °C, including a chaperone protein named DnaJ [51]. DnaJ is a heat shock protein also known as Hsp40 which coordinates with other heat shock proteins such as DnaK and GrpE in the folding of polypeptides and several other cellular functions [52]. A temperature-responsive signal transduction system called *RacR-RacS* was also identified in response to temperature changes with *C. jejuni* [53]. The two component regulators *RacR-RacS* consists of cytoplasmic sensory histidine kinase and a cytoplasmic membrane response regulator [26]. In most bacteria, this two component regulator system acts as a survival mechanism to protect against environmental changes by affecting the expression of stress response genes [54].

### 4.3. Refrigeration and Freezing Stress

Historically, *C. jejuni* is not able to survive in cold environments and this inability may be explained by the lack of cold shock proteins [38]. However, the ability of *Campylobacter* to survive at 4 °C or colder has been reported by various researchers. Survival of *C. jejuni* on raw chicken skin fragments at freezing temperatures (−20 °C) for 14 days and −70 °C for 56 days was reported by Lee et al. [55]. The ability to survive in cold temperature varies due to strain. Chan and others (2001) reported that strains that were isolated from humans can better survive at 4 °C compared to poultry isolates [56]. Similarly, some strains survive better at 4 °C than at 30 °C at different pHs and NaCl concentrations [57].

*C. jejuni* when incubated at 5 °C, genes involved in energy metabolism, particularly tricarboxylic acid cycle, oxidative phosphorylation, glycolysis and gluconeogenesis were found upregulated [58]. This indicates that *C. jejuni* utilizes energy for metabolism and survival in refrigeration temperature. Similarly, the superoxide dismutase gene (SOD) provides resistance to freeze–thaw that is induced by oxidative damage in *C. coli* [59]. In the same study, it was found that the SOD mutant was more sensitive to freezing and thawing compared to wild type.

The ability of *C. jejuni* to tolerate refrigeration and freezing temperatures is a food safety and public health challenge. However, *C. jejuni*’s mechanism for survival when exposed to cold temperatures has not been elucidated. Although this bacterium lacks cold shock proteins to tolerate and adapt to cold environments, it has still been isolated from chicken meat that was stored at 4 °C. Since most of the research has used single strains to understand the mechanism of *C. jejuni* survival, more comprehensive studies involving different strains are needed to fully understand the pattern of cold resistance, and and further genetic approaches should be explored to understand the mechanism of survival in cold environments.

### 4.4. Osmotic Stress

Osmotic homoeostasis is important for proper cell growth and maintenance. Slight imbalances in osmotic pressure or turgor pressure can disrupt the physiological balance of a cell and induce a VBNC state [26]. For Gram-negative bacteria, the minimum solute concentration in the cytoplasm that is necessary for growth is around 300 milliosmole [60]. Sodium chloride (NaCl) is the most common food preservative used and *C. jejuni* may encounter changes in osmolarity during food processing. While the role of osmotic stress is not fully understood, it has been reported that *C. jejuni* is sensitive to NaCl. *C. jejuni* was incapable of surviving in Mueller- Hinton (MH) medium that was supplemented with 2% NaCl but was able to survive in the presence of 0.5% to 1.5% NaCl [61]. In contrast, 44.6% (54/121) of *C. jejuni* (clinical strains) survived 2% NaCl, and 35.5% (43/121) of the strains were tolerant, even in 4% NaCl [13]. This indicates that osmotic tolerance might be strain specific. *C. jejuni* survive longer in the presence of 4.5% NaCl (*w*/*v*) in 4 °C environment when compared to 42 °C [61]. Similarly, no substantial decrease in *C.jejuni* was seen in 1.5% NaCl-treated minced poultry meat stored at 4 °C [62]. Likewise, *C. jejuni* was found to be able to survive for up to 96 h in non-growth temperatures (25 °C & 4 °C) in high osmolality (>175 mosmol) media [63]. This shows that *C. jejuni* ability to tolerate and grow in the presence of salt is mostly determined by temperature.

Several Gram-negative bacteria, including *E. coli* contain a high functioning potassium transporter system that responds to osmotic stress. However, *C. jejuni* lacks this osmotic change adaptation mechanism. It lacks the osmoregulatory betaine and trehalose biosynthetic pathways [64] observed in other bacteria, hence they cannot synthesize suitable solutes. Therefore, they must rely on other mechanisms to survive during osmotic changes. Filamentation has occurred in *C. jejuni* in response to hyper-osmotic stress [65]. Filamentous bacteria have historically been thought of as abnormal, but evidence suggests that they also play an important role in bacterial survival and pathogenicity [66].

### 4.5. UV Stress

Enteric pathogen survival can be affected by light. Ultraviolet radiation (UVR) is emitted from wavelengths of 10–400 nm and is subdivided into UV-A (315–400 nm), UV-B (280–315 nm), and UV-C (100–280 nm) [67]. The FDA has approved UV-C (with 90% emission at 253.7 nm) as a method of controlling surface bacteria on food products [68]. Research had been carried out to study the effect of natural sunlight on *C. jejuni* survival in sea and river water. It was found that *C. jejuni* was sensitive to sunlight with a high inactivation rate [69]. The effect of UV-C irradiation was studied in *E. coli*, *Yersinia* and *Campylobacter* (human clinical isolates), and the application of UV-C resulted in a 3 log CFU/mL reduction in all bacteria with a dosage of at least 1.8 mWs/cm^2^ required for *Campylobacter* reduction In vitro [70]. *C. jejuni* was inactivated by high intensity blue light (405 nm) on micro-well plate, which was more effective towards *C. jejuni* than *Salmonella* Enteritidis and *E. coli* O157: H7 [71]. While doing predictive modeling study, combination of different wavelengths of UV light particularly 280 and 300 nm was found to be effective in inactivating *C. jejuni* in vitro [72], whereas when using chicken meat (breast fillet) inoculated with *C. jejuni* instead of artificial growth media and exposing to UV treatment (0.192 J/cm^2^) resulted in 0.76 log CFU/g reduction [73]. In addition, UV-C reduced *C. jejuni* initially inoculated between 6.4–7.5 CFU/mL on broiler meat and skin by 0.7 log and 0.8 log, respectively, at 32.9 mWs/cm^2^ [74]. This demonstrates that UV is efficient against *Campylobacter* in vitro; however, when it comes to chicken meat, the opacity of the meat could prevent the penetration of UV light, hence reducing the effectiveness of the treatment. Additionally, the natural populations of *C. jejuni* were found to be more UV-resistant than culture collection strains, indicating that natural populations could be better equipped for stress survival [75].

UV disinfection has shown to induce VBNC in *E. coli*, *P. aeruginosa* and *S. aureus* [76,77]. Unfortunately, no study has analyzed if the UV light can induce VBNC in *C. jejuni*. Likewise, very limited study has analyzed the effect of UV light against *C. jejuni* in genetic level. RecA, a protein that helps DNA repair, was found to be expressed higher in *C. jejuni* after it was exposed to UV light, which suggests *C. jejuni* could be resistant to UV light [78].

### 4.6. Acid Stress

The ideal pH range for *C. jejuni* growth is 6.5 to 7.5 [23]. However, it has been found that *C. jejuni* (strain NCTC 11168) remained viable after 20 min exposure to pH 4.5 [79]. *Campylobacter* is sensitive to several organic acids. Exposure to 1% lactic acid for 5 min in broth at low temperature (5 °C) reduced *C. jejuni* by 2.1 log CFU/mL [80]. Organic acids such as acetic acid, citric acid, lactic acid, tartaric acid and malic acid of 0.5% final concentration decreased *C. jejuni* populations in broth (chicken juice and brain heart infusion broth) by 4 to 6 log CFU/mL (after 24 h); tartaric acid was the most effective [81]. In contrast to studies with broth, enteric pathogens like *Salmonella* and *Campylobacter* are protected from inactivation in extreme acid conditions (pH 2.5) when they are in food [82]. Foodborne pathogens may be more tolerable of acid stress in foods due to the widespread use of low pH settings for food preservation. On-farm therapies such as acidification of drinking water with organic acid, acidification of litter, and inclusion of feed additives (2 percent formic acid) are routinely used to reduce campylobacter colonization in poultry. Similarly, antimicrobials such as peracetic acid (PAA) and cetylpyridinium chloride (CPCL) are applied in chiller tanks and post-chill rinses to minimize the amount of *Campylobacter* and other food-borne pathogens in the processing plant. Continuous exposure of *Campylobacter* to low doses of acids may induce an adaptive tolerance response in *Campylobacter*. For example, a sublethal dose of stress provides protection towards subsequent lethal dose. This is shown in a study in which *C. jejuni* was exposed to pH 5.5 prior to challenge with a lethal dose of 4.5, it enabled 100-fold greater survival when compared to uninduced culture [83,84].

The ability of *C. jejuni* to overcome acid stress still needs to be explored. A new epidemiological pathway has been proposed for *C. jejuni* to survive a harsh environment. Protozoa have served as a natural vehicle to disseminate pathogens, such as *Helicobacter pylori* [85]. Protozoa are abundant in water systems. When co-cultured with *Acanthamoeba polyphaga*, *C. jejuni* were able to survive pHs well below their typical range, surviving for 20 h at pH 4 and 5 h at pH 2 [86]. This shows that a host can protect in severe environments and be a contributing factor to *C. jejuni* infections.

Certain genes, such as the *clpB* gene contribute to acid resistance [79]. Similarly, thioredoxin-disulfide (*TrxB*) and 19 KDa periplasmic protein (*p9*) genes were upregulated when *C. jejuni* was exposed to hydrochloric acid (HCl) and acetic acid [87]. Exposure to HCl or acetic acid also increased the expression of oxidative stress defense genes, such as *dps*, *sodB*, *trxB*, and *ahpC* in *C. jejuni* [87]. The Ferric uptake regulator gene (*Fur*) is important to the survival of *C. jejuni* when exposed to acid. *C. jejuni fur* mutant was more sensitive to acid than wild type, which shows that Fur is an important regulator during acid response [88].

## 5. Potential Survival Mechanisms

As previously mentioned, *C. jejuni* has confirmed to thrive several extreme environmental stressors like heat stress, acid stress and aerobic stress. However, how it manages to overcome stressors is still a question. Some of the potential survival meachnisms *C. jejuni* has been adopting to counteract environmental stressors has been discussed below.

### 5.1. Aerotolerance Development

The development of aerotolerance is one of the key documented mechanisms that allows *Campylobacter* to survive in harsh environments. *Campylobacter* strains linked to human illnesses have been found to exhibit significantly greater aerotolerance [13]. Different levels of aerotolerance have been used to describe the survivability of *Campylobacter* in oxygen rich environments [13,14,89]. Strains that survive for more than 24 h of oxygen exposure are referred to as Hyper aerotolerant strains and it’s prevalence is higher in both retail chicken and duck meat [47] as well as in humans [13]. Bacterial growth in the presence of oxygen produces ROS that are detoxified by the oxidative stress defense system. Unlike aerobes, microaerophiles and anaerobes exist in low-oxygen environments, yet have conserved oxidative stress resistance systems. The most common ROS-detoxification enzymes are alkyl hydroperoxide reductase, catalase, and superoxide dismutase. *C. jejuni* has single gene copies of alkyl hydroperoxide reductase, catalase, and superoxide dismutase (*ahpC*, *katA* and *sodB*) that play important roles in survival and morphological changes of *C. jejuni* under aerobic conditions [90]. Research shows that *C. jejuni* with mutation in key antioxidant genes, including *ahpC*, *katA*, and *sodB* exhibited growth reduction under aerobic conditions [11]. Proteins including alkyl hydroperoxide reductase AhpC [11], ferredoxin FdxA, protease HtrA, and a shortened hemoglobin Ctb expressed are also involved in oxygen tolerance in *C. jejuni* [91,92,93,94]. Previously annotated as a hypothetical protein, Cj1556 was discovered as a MarR family transcriptional regulator after reannotation of the *C. jejuni* NCTC11168 genome sequence, and additional investigation revealed a potential involvement in regulating the oxidative stress response [95] Increased sensitivity to oxidative and aerobic stress was observed in a *C. jejuni* 11168H transcriptional regulator Cj1556 mutant which demonstrated that transcriptional regulator Cj1556 has role in the regulation of aerobic stress [95]. Thiol peroxidase genes, *tpx* and *bcpb* encode for enzymes Tpx and Bcp. These enzymes demonstrated a role in regulating oxidative stress responses in *C. jejuni*, most specially those related to molecular oxygen [96]. Aerotolerance and oxidative stress response may be closely related given that there are common genes expressed in both of these survival mechanisms (Table 3).

### 5.2. Biofilm Formation

Biofilms are bacterial communities, which are encased in a matrix that are irreversibly adhered to one another and/or to surfaces or interfaces [97]. Biofilms help bacteria survive by protecting them from environmental stresses such as dehydration and antibacterial and sanitizing agents, which makes their elimination difficult. Biofilms created on food processing surfaces shield bacteria from washing and sanitation procedures, which can lead to food contamination and food borne illnesses. It is believed that biofilm formation is one of the mechanisms contributing towards *C. jejuni* persistence in the environment [98]. While most investigations of *C. jejuni* biofilm development have been conducted under microaerobic circumstances, limited research have been conducted to investigate *C. jejuni* biofilm production under aerobic conditions. *C. jejuni* form a stronger biofilm under lower oxygen conditions [17]. Likewise, *C. jejuni* showed enhanced biofilm formation by use of Fe^2+^ or Fe^3+^ or both [99]. Iron has shown to enhances the biofilm formation of *C. jejuni* through increasing oxidative stress. The iron molecules may create a microaerobic environment that is conducive to their life and development. More research on biofilm development by *C. jejuni* is needed since many aspects remain unclear, especially *C. jejuni*’s capacity to build biofilms under environmental oxidative stress.

*C. jejuni* forms biofilms on stainless steel, nitrocellulose, and glass fiber filters [100]. The ability to form biofilm is enhanced when it is linked with food surfaces or organic components in food. Chicken meat exudate increases *C. jejuni* biofilm formation on glass, polystyrene, and stainless steel when supplemented with brucella broth [19]. Interestingly, *Campylobacter* isolated from retail food samples was able to form more biofilm in co-culture with *E. coli* or *Pseudomonas aeruginosa* than in pure culture [101]. Similar results occurred when *C. jejuni* multilocus sequence type ST-47, a dominant poultry and human-associated type in New Zealand, exhibited greater biofilm formation when grown in mixed microbial population with *Enterococcus faecalis* and *Staphylococcus simulans*. It could be because these microorganisms derived from chicken may create a favorable habitat for *C. jejuni* survival and proliferation in poultry processing plant settings.

*C. jejuni* has a minimum growth temperature requirement of 37 °C to 42 °C [102,103]. The failure to replicate and develop below 30 °C may be explained in part by a lack of stress adaptive response components, such as cold stress proteins. Nonetheless, *C. jejuni* is metabolically active at temperatures lower than its lowest growth temperature. It has been shown that survival of planktonic *C. jejuni* is improved at low temperature (4 °C) [56]. Moreover, *C. jejuni* cultured as planktonic cells and biofilm cells lived longer at lower temperatures (4 °C and 10 °C) than at higher temperatures (25 °C and 37 °C) [22]. *C. jejuni*’s ability to build biofilms may alter depending on temperature. However, research on the influence of temperature on *C. jejuni* biofilm production is sparse. A greater knowledge of the processes involved in biofilm development is critical in establishing cleaning procedures, which will reduce *Campylobacter* persistence and transmission. Strategies to eradicate *Campylobacter* reservoirs during food manufacturing would benefit our understanding of how to limit the incidence and transmission of Campylobacteriosis and other food-borne diseases throughout the food chain.

#### Genes Involved in Biofilm Formation by *Campylobacter*

*Campylobacter* strains differ in their ability to form biofilms [104], which impact their ability to survive outside of the host, as well as transmission and colonization of various host species [105]. Genetic determinants of biofilm formation vary between species. However, strains of the same species with variation in homologous regions differ in biofilm phenotype [18]. The ability of 102 *C. jejuni* isolates to form biofilms and the associated genetic factors were investigated by Pasoce et al. [18]. They reported a total of 46 biofilm-related genes, including those involved in adhesion, motility, glycosylation, capsule formation, and oxidative stress. They found four genes, *trxA*, *trxB*, *ilvE*, and *nuoC*, associated with oxidative stress sensing, which is strongly linked to biofilm formation. Studies show that genes involved in motility (*flgA*) [106], quorum sensing (luxS) [17] and adhesion (*cadF*) and stress response genes such as *dnaJ*, *cbrA*, *htrA*, and *sodB* [107] are involved in biofilm formation. Similar to this, *C. jejuni* mutant of *AhpC* gene, which is involved in oxidative stress, had greater biofilm formation [107]. Cell binding proteins, Peb1A and Peb4 (also known as CBF1 and CBF2), play important roles in adhesion and in vivo colonization of *Campylobacter* [108]. Deletion of the *peb4* gene impaired cell adhesion and biofilm formation [109]. Briefly, the list of potential genes involved in *C. jejuni* biofilm formation along with their functions are in Table 4. *C. jejuni* induce biofilm formation by activation of various stress regulators [109]. *Campylobacter*’s ability to form biofilms is dependent upon strain, and *Campylobacter* forms multispecies biofilm with other bacteria to thrive in harsh environments.

Several strategies have been evaluated to target *Campylobacter* biofilms. Extra cellular polymeric substance produced by the bacteria is an essential component of biofilm and it is composed of extracellular DNA (eDNA) [110]. Use of DNase can degrade the eDNA and provide a promising measure to inhibit biofilm [111]. Zinc nano particles (0.5 mg/mL) reduces *C. jejuni* biofilm formation by both in mono and multispecies culture under aerobic condition [101]. Natural compounds, such as carvacrol (66.56 mM), attenuate *C. jejuni* colonization and biofilm mass by greater than 7 log CFU/mL reduction [112]. Similarly, plant derived compounds like citrus extract at the concentration of 75% of minimum bactericidal concentration of 130–250 µg/mL [113], cinnamaldehyde at concentration greater than 15.63 µg/mL [114], eugenol at concentration of 60.9 mM [115] has shown promising effect in decreasing *Campylobacter* biofilms. In summary, several strategies to control *Campylobacter* biofilms have been studied but the practical aspect of using them in food industry needs to be further investigated. Some treatments are successful in the lab when bacteria are cultivated in culture plates, and most treatments are focused on a single strain of *Campylobacter* spp. However, the substantial genetic diversity among *Campylobacter* strains complicates treatment effectiveness on food contact surfaces in food plants.

### 5.3. Viable but non Culturable (VBNC) State Formation

Rollins and Colwell (1986) first described VBNC state in *C. jejuni* when they found that increasing cultivation temperature resulted in decreased spread plate culture counts when compared to direct viable counts with epifluorescence microscopy and assaying protein synthesis [28]. Prolonged incubation of *C. jejuni* at 4 °C leads to the formation of VBNC state which retain the ability to invade CaCo-2 human intestinal epithelial cells in vitro as reported by Chaisowwong et al. [29]. This might explain why, despite better sanitation and food preparation standards, campylobacteriosis infections and outbreaks are still common in developed nations.

Several genes are involved in VBNC state formation in *C. jejuni* (Table 5). Transcription of virulence genes, including *cadF*, *ciaB*, *flaA*, *flaB*, *cdtA*, *cdtB*, and *cdtC*, were down regulated in *C. jejuni* during the VBNC state [29]. An outer-membrane protein (CadF) expression was found to retain the ability of VBNC *C. jejuni* to adhere to CaCo-2 cells [116]. During the VBNC state (when incubated at 5 °C), gene expression involved in energy metabolism increased [58], which indicates that they conserve energy by decreasing their virulence activity and utilize energy for metabolism and survival.

The VBNC form of *C. jejuni* can resuscitate [24,117,118,119]. Non culturable *Campylobacter* in water sources were able to colonize in hens following water intake [120]. There are several other studies that shows the ability of *C. jejuni* to resuscitate in laboratory animals [24,117,118,119]. In contrast, oral inoculation of laboratory animals and volunteers with non-culturable *C. jejuni* did not show any symptoms and no campylobacter was detected in the stool [121]. Similarly, *C. jejuni* was not isolated from the caeca of chicks 14d after oral inoculation of non-culturable *C. jejuni* [102]. This suggests that the resuscitation mechanism might differ between strains, hosts, and/or transmission vector. Further investigation is needed to understand the possible mechanisms fully for *C. jejuni* to shift from the non-culturable state to the culturable state.

*C. jejuni* passes through different levels of environmental stressors when a bird is processed. Since shifting to the VBNC state is a response to an unfavorable environment, *C. jeuni* could shift to the non-culturable state after passing through all stressors, but remain in the chicken. Viable cells were detected up to 7 months later when incubated at 4 °C [122]. Studies have been reported that *C. jejuni* can form VBNC due to acid stress [23] and cold stress [29]. However, the VBNC forming ability of *Campylobacter* after passing through stressors, and its ability to cause infection in humans is still not clear. Therefore, there is a need to understand the mechanism underlying environmental stress resistance in *Campylobacter* that leads to the transition to the VBNC state.

## 6. Future Research to Fill Current Knowledge Gaps

Researchers have predominantly addressed the effect of various stressors such as aerobic stress, acidity, osmotic imbalance, freeze–thaw, high temperatures, and UV stress at the phenotypic level [11,94] and identified genes that are associated with different phenotypes. However, limited research has been conducted to determine the expression level of those genes when exposed to stress. The focus of future study should be on understanding the underlying adaptive responses, which may be accomplished by gene expression profiling.

Furthermore, understanding the ability to shift from a non-culturable to a culturable state, as well as the substantial genetic diversity among *Campylobacter* strains (evolutionary adaptation), may aid in the discovery of additional pathways involved in stress resistance, which may ultimately lead to the development of effective measures to control the risk of *Campylobacter* in the food supply chain. The biofilm-forming characteristics of *Campylobacter* are still unknown compared to other foodborne pathogens such as *Salmonella*. Thus, knowing genetic diversity and its involvement in biofilm development is critical to the removal of *Campylobacter* from food processing and manufacturing plants.

## 7. Conclusions

While there have been a number of technological food safety advances in chicken meat processing, *Campylobacter* remains the most common pathogen found in poultry meat, which results in foodborne illnesses and outbreaks in the United States. There are a variety of interventions that commercial processing facilities employ to decrease the incidence of *Campylobacter* in retail food products. Use of hot water treatments (scalding), chilling and freezing, as well as use of antimicrobials are the most utilized treatments in the processing plant. However, *Campylobacter* still poses a threat to food safety (Table 1) and public health due to its ability to adapt to stress, form biofilms, and shift to the VNBC state.

**Table 1 microorganisms-10-02165-t001:** Incidence of *Campylobacter jejuni* at various stages inside processing plants.

Source of Contamination	Country of Occurrence	Incidence of *C. jejuni*	References
Birds with feathers, birds at rehang, birds after evisceration, immediately after entering chiller, after exiting chlorinated chill tank	USA—Georgia	87.27%, (*n* = 55)	[123]
Rehang and post chill whole carcass rinse	USA—Alabama, Arkansas, California, Delaware, Georgia, Indiana, Missouri, North Carolina, South Carolina, Tennessee, Texas, Virginia, and West Virginia	Rehang—74.5%, (*n* = 800) Post-chill—34.90%, (*n* = 800)	[124]
Feces, pasture soil, whole carcass rinse directly after processing (WCR-P), final product whole carcass rinse after chilling and storage time (WCR-F), and ceca samples collected during processing from each farm	Southeastern USA from March 2014 to November 2017	39.08%, (*n* = 2305)	[125]
Air, feces-litter, feed pans and water lines	USA—Virginia	26.66%, (*n* = 120)	[126]
Fecal and environmental samples	USA—North Carolina	Fecal: 29.50%; (*n* = 400) Environmental sample: 1%, (*n* = 500)	[127]
Pre-scald (feather and skin)	USA—Delmarva Peninsula	77%, (*n* = 48)	[128]
Evisceration	USA	96–100%, (*n* = 48)	[129]
Slaughterhouse	Brazil- Parana, santa Catarina and Rio Grande do Sul	*Campylobacter* spp.—35.84% (*n* = 816) *C. jejuni*—78.47% (*n* = 144)	[130]
Defeathering, Evisceration, Shackles, Converyor belt	North of Spain	Defeathering—80% (*n* = 30) Eviseration—100% (*n* = 39) Shackles—100% (*n* = 23) Converyor belt—96.6% (*n* = 29)	[131]
Defeathering machine, evisceration machine, conveyor belts, scald tank, water	France	87% (34/39)	[45]

*C. jejuni* utilizes numerous mechanisms to adapt to environmental stressors (Table 2 and Table 3). *C. jejuni* induces biofilm formation (Table 4) in some strains by activation of various stress regulators. There have been several control strategies to remove biofilms from processing facilities. However, there is substantial genetic diversity among campylobacter strains, which complicates treatment effectiveness.

**Table 2 microorganisms-10-02165-t002:** Cluster of genes and their potential functions in overcoming multiple stress that *C. jejuni* encounters in processing facility.

Stressors	Gene Involved	Gene Function	References
Heat stress	(*i*) *dnaJ*	Express DnaJ chaperone protein-protein folding and heat shock response	[51]
	(*ii*) *racR*	Component of RacR-RacS temperature-responsive signal transduction system	[53]
	(*iii*) *htrA*	High temperature requirement A (HtrA)-like protease and chaperones	[91]
	(*iv*) *dps*	Express iron sequestration ferritin protein (Dps)	[132]
	(*v*) *dnaK*	Express DnaK chaperonin- protein folding	
Refrigeration and freezing stress	(*i*) *sodB*	Superoxide dismutase	[59]
	(*ii*) *luxS*	S-ribosylhomocysteinase-catalyzes the formation of autoinducer-2 (AI-2) molecules and homocysteine	[133]
Osmotic stress	(*i*) *kpsM*	Capsule export gene-encodes capsule export apparatus	[65]
Acid stress	(*i*) *clpB*	ATP-dependent Protease-heat shock gene	[79]
	(*ii*) *trxB*	Thioredoxin-disulfide reductase-protein folding	[87]
	(*iii*) *fur*	Ferric uptake regulator gene	[88]
UV stress	*recA*	Express recA protein involved in DNA repair	[78]

**Table 3 microorganisms-10-02165-t003:** Cluster of genes involved, and its potential function in aerotolerance mechanism in *Campylobacter jejuni*.

Gene	Gene Product or Function	Involved Function in Aerotolerance	References
*ahpC*	Alkyl hydroperoxide reductase-Antioxidant	Antioxidant	[40,90]
*katA*	Catalase	Peroxide detoxification	[90]
*sodB*	Iron co-factored superoxide dismutase-	Antioxidant	[90]
*fdxA*	Ferredoxin A-cyclophilin gene	Antioxidant	[93]
*htrA*	High temperature requirement-A protease	Removal of misfolded proteins as a result of oxygen stress	[92]
*tpx*	Thiol peroxidase-	Scavenges molecular oxygen	[96]
*bcpb*	Express Bacterioferritin comigratory protein	Regulate oxidative stress by attacking molecular oxygen	[96]
*ctb*	Truncated hemoglobin	Oxygen-protective physiological role by increasing oxygen uptake rates	[94]
Cj1556	MarR family transcriptional regulator	Oxidative stress response	[95]

**Table 4 microorganisms-10-02165-t004:** Cluster of genes involved and its function in biofilm formation in *Campylobacter jejuni*.

Gene	Gene Product or Function	Involved Function in Biofilm	References
*flaA*	Glycosylated structural flagellins-A	Involved in cell associated with motility	[106]
*luxS*	S-ribosylhomocysteianse	Quorum sensing	[17]
*cadF*	*Campylobacter* adhesion to fibronectin	Adhesion	[17]
*dnaJ*	Chaperone DnaJ	Stress response	[107]
*cbrA*	*Campylobacter* bile resistance regulator	Stress response	[107]
*htrA*	High temperature requirement A	Stress response	[107]
*sodB*	Superoxide dismutase	Stress response	[107]
*ahpC*	Alkyl hydroperoxide reductase	Involved in oxidative stress response	[107]
*peb4*	encode homolog of cluster 3 binding protein	adhesion	[109]
*trxA*	Thioredoxin A	Involved in oxidative stress response	[18]
*trxB*	Thioredoxin B	Involved in oxidative stress response	[18]
*ilvE*	Branched chain amino transferases for leucine, isoleucine, and valine	Involved in oxidative stress response	[18]
*nuoC*	NuoC- subunit of complex I (ubiquinone oxidoreductase)	Assembly or the stability of ubiquinone oxidoreductase	[18]

VBNC state is another response mechanism to unfavorable environment condition (Table 5). *C. jejuni* cand shift to the non-culturable state after passing through stressors and remain in food, with a potential to cause human infection. However, the mechanism underlying this adaptive response is still not clear. Further research needs to be conducted to understand the genetic aspect of the survival mechanism so that effective control measures can be developed.

**Table 5 microorganisms-10-02165-t005:** Cluster of genes involved and its function in VBNC formation in *Campylobacter jejuni*.

Gene	Gene Product or Function	Involved Function in VBNC	References
*ppk1*	Poly phosphate kinase 1-codes for PPK1 enzyme mediates the synthesis of poly P	Mutant of *ppk1* decreased the accumulation of poly P-decreased stress response	[134]
*cadF*	Code 37 kDa adhesin-bind to fibronectin and mediates bacteria-host interaction	Retain ability to adhere to host cells	[116]
*flaA*, *flab*	Flagellin-Involved bacteria internalization	Decreased expression- conserve energy for metabolism	[29]
*cdtA*, *cdtB* and *cdtC*	Cytolethal distending toxins-arrest G2/M phase of cell cycle causing cell death	Decreased expression-conserve energy for metabolism	[29]
*ciaB*	*Campylobacter* invasion antigen B	Decreased expression-conserve energy for metabolism	[29]

## Figures and Tables

**Figure 1 microorganisms-10-02165-f001:**
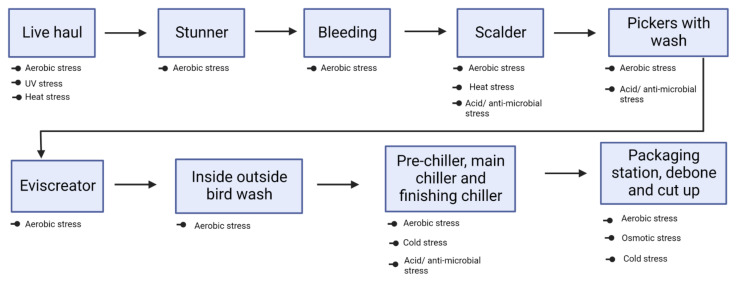
Potential stressors encountered by *C. jejuni* at various stages of broiler processing plant.

## Data Availability

No new data were created or analyzed in this study. Data sharing is not applicable to this article.

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
