# Peer review of "Roles of Aerotolerance, Biofilm Formation, and Viable but Non-Culturable State in the Survival of Campylobacter jejuni in Poultry Processing Environments"

_microorganisms, 2022, doi:10.3390/microorganisms10112165_

Round 1

Reviewer 1 Report

The review by Pokhrel co-workers aims to report an overview on the different stressors that C. jujuni may encounter in the production chain.

The review is scientifically sound and reports data and experiences previously published about the general survival mechanisms.

The review is not a novelty and some other similar publications exist.

The first point that I would like to point out is the missing of any figure to guide the readers in the main topics reported in the publication.

I think it would be of great help to have a main figure to explain the main stressors and the main responses.

I also think that a general review need to also report general data (number cases reported, incidence in poultry processing facilities) not only focused in US but having a broader range including also other geographical areas. I’m sure data are available from Europe, Australia, New Zealand. Please update the statements.

In the flow of the review I found a little bit confusing to describe first the stresses encountered by C. jejuni in a potential poultry processing facility and then the survival mechnisms describing the aerotollerance, biofilms and VBNC states.

In the stresses I think is missing the desiccation and relating it with the osmotic stress.

Moreover I think here is missing a paragraph for detailing a general adaptive response that may or not be related to the afore mentioned mechanisms Something is reported in the previous paragraphs but the readers needs to understand if there are other mechanisms and association with strains capabilities.

Saying that I suggest to revise the review adding some self-explaining figure with stress and point of occurrence in the facility. Another figure highlighting the survival mechanisms in the processing plat may be also very useful.

I also suggest to report data also from other Countries out of US.

Author Response

Reviewer 1

The review by Pokhrel co-workers aims to report an overview on the different stressors that C. jujuni may encounter in the production chain.

The review is scientifically sound and reports data and experiences previously published about the general survival mechanisms.

The review is not a novelty and some other similar publications exist.

The first point that I would like to point out is the missing of any figure to guide the readers in the main topics reported in the publication.

We thank the reviewer for this comment. We have added a figure in line 879 of the revised manuscript to summarize the stressors encountered by C. jejuni in broiler processing. Moreover, we have five tables summarizing different types of data.

I think it would be of great help to have a main figure to explain the main stressors and the main responses.

We thank the reviewer for this comment. Like mentioned in the previous comment, we have added a new figure.

I also think that a general review need to also report general data (number cases reported, incidence in poultry processing facilities) not only focused in US but having a broader range including also other geographical areas. I’m sure data are available from Europe, Australia, New Zealand. Please update the statements.

We thank the reviewer for this comment. Data from outside the US has been added in a table in line 873.

In the flow of the review I found a little bit confusing to describe first the stresses encountered by C. jejuni in a potential poultry processing facility and then the survival mechnisms describing the aerotollerance, biofilms and VBNC states.

We thank the reviewer for this comment. A connecting statement has been included in line 328 to improve the flow of paper.

In the stresses I think is missing the desiccation and relating it with the osmotic stress.

We thank the reviewer for this comment. Again, there are several other stressors that are not explained in this review paper but have been discussed in detail in other review papers like [1] . We have only discussed some of the major stressors encountered in different stages of poultry processing.

Moreover I think here is missing a paragraph for detailing a general adaptive response that may or not be related to the afore mentioned mechanisms Something is reported in the previous paragraphs but the readers needs to understand if there are other mechanisms and association with strains capabilities.

We thank the reviewer for the comment. We agree that there are several other survival mechanisms that campylobacter use to overcome stressors. The general survival mechanisms have already been discussed in paper [1]. The main focus of this paper is to discuss in depth of Biofilm, VBNC formation and aerotolerance.

Saying that I suggest to revise the review adding some self-explaining figure with stress and point of occurrence in the facility. Another figure highlighting the survival mechanisms in the processing plat may be also very useful.

We thank the reviewer for this comment. A figure has been added in line 879 to highlight stress and point of occurrence.

I also suggest to report data also from other Countries out of US.

Thank you for this comment. International data have been added in table in line 873.

Reviewer 2

This review is timely and addresses and important topic. It will be useful for both researchers and industry. It covers a wide and balanced range of literature sources and addresses all topics fairly. Aside from some very minor additions or changes I feel this is a very good review.

We thank the reviewer for reading the supplied manuscript. Based on comments from the reviewers several changes were made in the addressed comments above to enhance the readability of the publication.

Reviewer 2 Report

This review is timely and addresses and important topic. It will be useful for both researchers and industry. It covers a wide and balanced range of literature sources and addresses all topics fairly. Aside from some very minor additions or changes I feel this is a very good review.

Author Response

(The authors gave the same response as above.)
